# PREDICT THEN PROPAGATE: GRAPH NEURAL NETWORKS MEET PERSONALIZED PAGERANK

**Johannes Gasteiger, Aleksandar Bojchevski & Stephan Günnemann**
Technical University of Munich, Germany
{j.gasteiger,a.bojchevski,guennemann}@in.tum.de

## ABSTRACT

Neural message passing algorithms for semi-supervised classification on graphs have recently achieved great success. However, for classifying a node these methods only consider nodes that are a few propagation steps away and the size of this utilized neighborhood is hard to extend. In this paper, we use the relationship between graph convolutional networks (GCN) and PageRank to derive an improved propagation scheme based on personalized PageRank. We utilize this propagation procedure to construct a simple model, personalized propagation of neural predictions (PPNP), and its fast approximation, APPNP. Our model's training time is on par or faster and its number of parameters on par or lower than previous models. It leverages a large, adjustable neighborhood for classification and can be easily combined with any neural network. We show that this model outperforms several recently proposed methods for semi-supervised classification in the most thorough study done so far for GCN-like models. Our implementation is available online. [1]

## 1  INTRODUCTION

Graphs are ubiquitous in the real world and its description through scientific models. They are used to study the spread of information, to optimize delivery, to recommend new books, to suggest friends, or to find a party's potential voters. Deep learning approaches have achieved great success on many important graph problems such as link prediction (Grover & Leskovec, 2016; Bojchevski et al., 2018), graph classification (Duvenaud et al., 2015; Niepert et al., 2016; Gilmer et al., 2017) and semi-supervised node classification (Yang et al., 2016; Kipf & Welling, 2017).

There are many approaches for leveraging deep learning algorithms on graphs. Node embedding methods use random walks or matrix factorization to directly train individual node embeddings, often without using node features and usually in an unsupervised manner, i.e. without leveraging node classes (Perozzi et al., 2014; Tang et al., 2015; Nandanwar & Murty, 2016; Grover & Leskovec, 2016; Qiu et al., 2018). Many other approaches use both graph structure and node features in a supervised setting. Examples for these include spectral graph convolutional neural networks (Bruna et al., 2014; Defferrard et al., 2016), message passing (or neighbor aggregation) algorithms (Kearnes et al., 2016; Kipf & Welling, 2017; Hamilton et al., 2017; Pham et al., 2017; Monti et al., 2017; Gilmer et al., 2017), and neighbor aggregation via recurrent neural networks (Scarselli et al., 2009; Li et al., 2016; Dai et al., 2018). Among these categories, the class of message passing algorithms has garnered particular attention recently due to its flexibility and good performance.

Several works have been aimed at improving the basic neighborhood aggregation scheme by using attention mechanisms (Kearnes et al., 2016; Hamilton et al., 2017; Veličković et al., 2018), random walks (Abu-El-Haija et al., 2018a; Ying et al., 2018; Li et al., 2018), edge features (Kearnes et al., 2016; Gilmer et al., 2017; Schlichtkrull et al., 2018) and making it more scalable on large graphs (Chen et al., 2018; Ying et al., 2018). However, all of these methods only use the information of a very limited neighborhood for each node. A larger neighborhood would be desirable to provide the model with more information, especially for nodes in the periphery or in a sparsely labelled setting.

Increasing the size of the neighborhood used by these algorithms, i.e. their range, is not trivial since neighborhood aggregation in this scheme is essentially a type of Laplacian smoothing and too

---

many layers lead to oversmoothing (Li et al., 2018). Xu et al. (2018) highlighted the same problem by establishing a relationship between the message passing algorithm termed Graph Convolutional Network (GCN) by Kipf & Welling (2017) and a random walk. Using this relationship we see that GCN converges to this random walk's limit distribution as the number of layers increases. The limit distribution is a property of the graph as a whole and does not take the random walk's starting (root) node into account. As such it is unsuited to describe the root node's neighborhood. Hence, GCN's performance necessarily deteriorates for a high number of layers (or aggregation/propagation steps).

To solve this issue, in this paper, we first highlight the inherent connection between the limit distribution and PageRank (Page et al., 1998). We then propose an algorithm that utilizes a propagation scheme derived from *personalized* PageRank instead. This algorithm adds a chance of teleporting back to the root node, which ensures that the PageRank score encodes the local neighborhood for every root node (Page et al., 1998). The teleport probability allows us to balance the needs of preserving locality (i.e. staying close to the root node to avoid oversmoothing) and leveraging the information from a large neighborhood. We show that this propagation scheme permits the use of far more (in fact, infinitely many) propagation steps without leading to oversmoothing.

Moreover, while propagation and classification are inherently intertwined in message passing, our proposed algorithm *separates* the neural network from the propagation scheme. This allows us to achieve a much higher range without changing the neural network, whereas in the message passing scheme every additional propagation step would require an additional layer. It also permits the independent development of the propagation algorithm and the neural network generating predictions from node features. That is, we can combine any state-of-the-art prediction method with our propagation scheme. We even found that adding our propagation scheme during inference significantly improves the accuracy of networks that were trained without using any graph information.

Our model achieves state-of-the-art results while requiring fewer parameters and less training time compared to most competing models, with a computational complexity that is linear in the number of edges. We show these results in the most thorough study (including significance testing) of message passing models using graphs with text-based features that has been done so far.

## 2 GRAPH CONVOLUTIONAL NETWORKS AND THEIR LIMITED RANGE

We first introduce our notation and explain the problem our model solves. Let $G = (V, E)$ be a graph with nodes $V$ and edges $E$. Let $n$ denote the number of nodes and $m$ the number of edges. The nodes are described by the feature matrix $\boldsymbol{X} \in \mathbb{R}^{n \times f}$, with the number of features $f$ per node, and the class (or label) matrix $\boldsymbol{Y} \in \mathbb{R}^{n \times c}$, with the number of classes $c$. The graph $G$ is described by the adjacency matrix $\boldsymbol{A} \in \mathbb{R}^{n \times n}$. $\tilde{\boldsymbol{A}} = \boldsymbol{A} + \boldsymbol{I}_n$ denotes the adjacency matrix with added self-loops.

One simple and widely used message passing algorithm for semi-supervised classification is the Graph Convolutional Network (GCN). In the case of two message passing layers its equation is

$$\boldsymbol{Z}_{\text{GCN}} = \text{softmax}\left(\hat{\tilde{\boldsymbol{A}}} \, \text{ReLU}\left(\hat{\tilde{\boldsymbol{A}}} \boldsymbol{X} \boldsymbol{W}_0\right) \boldsymbol{W}_1\right), \tag{1}$$

where $\boldsymbol{Z} \in \mathbb{R}^{n \times c}$ are the predicted node labels, $\hat{\tilde{\boldsymbol{A}}} = \tilde{\boldsymbol{D}}^{-1/2} \tilde{\boldsymbol{A}} \tilde{\boldsymbol{D}}^{-1/2}$ is the symmetrically normalized adjacency matrix with self-loops, with the diagonal degree matrix $\tilde{\boldsymbol{D}}_{ij} = \sum_k \tilde{\boldsymbol{A}}_{ik} \delta_{ij}$, and $\boldsymbol{W}_0$ and $\boldsymbol{W}_1$ are trainable weight matrices (Kipf & Welling, 2017).

With two GCN-layers, only neighbors in the two-hop neighborhood are considered. There are essentially two reasons why a message passing algorithm like GCN cannot be trivially expanded to use a larger neighborhood. First, aggregation by averaging causes oversmoothing if too many layers are used. It, therefore, loses its focus on the local neighborhood (Li et al., 2018). Second, most common aggregation schemes use learnable weight matrices in each layer. Therefore, using a larger neighborhood necessarily increases the depth and number of learnable parameters of the neural network (the second aspect can be circumvented by using weight sharing, which is typically not the case, though). However, the required neighborhood size and neural network depth are two completely orthogonal aspects. This fixed relationship is a strong limitation and leads to bad compromises.

We will start by concentrating on the first issue. Xu et al. (2018) have shown that for a $k$-layer GCN the influence score of node $x$ on $y$, $I(x, y) = \sum_i \sum_j \frac{\partial \boldsymbol{Z}_{yi}}{\partial \boldsymbol{X}_{xj}}$, is proportional in

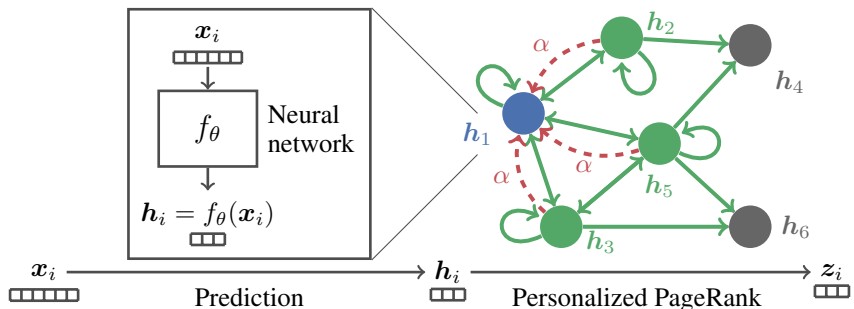

Figure 1: Illustration of (approximate) personalized propagation of neural predictions (PPNP, APPNP). Predictions are first generated from each node's own features by a neural network and then propagated using an adaptation of personalized PageRank. The model is trained end-to-end.

expectation to a slightly modified $k$-step random walk distribution starting at the root node $x$, $P_{\text{rw'}}(x \to y, k)$. Hence, the information of node $x$ spreads to node $y$ in a random walk-like manner. If we take the limit $k \to \infty$ and the graph is irreducible and aperiodic, this random walk probability distribution $P_{\text{rw'}}(x \to y, k)$ converges to the limit (or stationary) distribution $P_{\text{lim}}(\to y)$. This distribution can be obtained by solving the equation $\boldsymbol{\pi}_{\text{lim}} = \hat{\tilde{\boldsymbol{A}}}\boldsymbol{\pi}_{\text{lim}}$. Obviously, the result only depends on the graph as a whole and is independent of the random walk's starting (root) node $x$. This global property is therefore unsuitable for describing the root node's neighborhood.

## 3   PERSONALIZED PROPAGATION OF NEURAL PREDICTIONS

**From message passing to personalized PageRank.** We can solve the problem of lost focus by recognizing the connection between the limit distribution and PageRank (Page et al., 1998). The only differences between these two are the added self-loops and the adjacency matrix normalization in $\hat{\tilde{\boldsymbol{A}}}$. Original PageRank is calculated via $\boldsymbol{\pi}_{\text{pr}} = \boldsymbol{A}_{\text{rw}}\boldsymbol{\pi}_{\text{pr}}$, with $\boldsymbol{A}_{\text{rw}} = \boldsymbol{A}\boldsymbol{D}^{-1}$. Having made this connection we can now consider using a variant of PageRank that takes the root node into account – personalized PageRank (Page et al., 1998). We define the root node $x$ via the teleport vector $\boldsymbol{i}_x$, which is a one-hot indicator vector. Our adaptation of personalized PageRank can be obtained for node $x$ using the recurrent equation $\boldsymbol{\pi}_{\text{ppr}}(\boldsymbol{i}_x) = (1-\alpha)\hat{\tilde{\boldsymbol{A}}}\boldsymbol{\pi}_{\text{ppr}}(\boldsymbol{i}_x)+\alpha\boldsymbol{i}_x$, with the teleport (or restart) probability $\alpha \in (0, 1]$. By solving this equation, we obtain

$$\boldsymbol{\pi}_{\text{ppr}}(\boldsymbol{i}_x) = \alpha \left( \boldsymbol{I}_n - (1 - \alpha)\hat{\tilde{\boldsymbol{A}}} \right)^{-1} \boldsymbol{i}_x. \tag{2}$$

Introducing the teleport vector $\boldsymbol{i}_x$ allows us to preserve the node's local neighborhood even in the limit distribution. In this model the influence score of root node $x$ on node $y$, $I(x, y)$, is proportional to the $y$-th element of our personalized PageRank $\boldsymbol{\pi}_{\text{ppr}}(\boldsymbol{i}_x)$. This value is different for every root node. How fast it decreases as we move away from the root node can be adjusted via $\alpha$. By substituting the indicator vector $\boldsymbol{i}_x$ with the unit matrix $\boldsymbol{I}_n$ we obtain our fully personalized PageRank matrix $\boldsymbol{\Pi}_{\text{ppr}} = \alpha(\boldsymbol{I}_n - (1 - \alpha)\hat{\tilde{\boldsymbol{A}}})^{-1}$, whose element $(yx)$ specifies the influence score of node $x$ on node $y$, $I(x, y) \propto \boldsymbol{\Pi}_{\text{ppr}}^{(yx)}$. Note that due to symmetry $\boldsymbol{\Pi}_{\text{ppr}}^{(yx)} = \boldsymbol{\Pi}_{\text{ppr}}^{(xy)}$, i.e. the influence of $x$ on $y$ is equal to the influence of $y$ on $x$. This inverse always exists since $\frac{1}{1-\alpha} > 1$ and therefore cannot be an eigenvalue of $\hat{\tilde{\boldsymbol{A}}}$ (see Appendix A).

**Personalized propagation of neural predictions (PPNP).** To utilize the above influence scores for semi-supervised classification we generate predictions for each node based on its own features and then propagate them via our fully personalized PageRank scheme to generate the final predictions. This is the foundation of personalized propagation of neural predictions. PPNP's model equation is

$$\boldsymbol{Z}_{\text{PPNP}} = \text{softmax}\left( \alpha \left( \boldsymbol{I}_n - (1 - \alpha)\hat{\tilde{\boldsymbol{A}}} \right)^{-1} \boldsymbol{H} \right), \qquad \boldsymbol{H}_{i,:} = f_\theta(\boldsymbol{X}_{i,:}), \tag{3}$$

where $\boldsymbol{X}$ is the feature matrix and $f_\theta$ a neural network with parameter set $\theta$ generating the predictions $\boldsymbol{H} \in \mathbb{R}^{n \times c}$. Note that $f_\theta$ operates on each node's features independently, allowing for parallelization. Furthermore, one could substitute $\hat{\tilde{\boldsymbol{A}}}$ with any propagation matrix, such as $\boldsymbol{A}_{\text{rw}}$.

As a consequence, PPNP separates the neural network used for generating predictions from the propagation scheme. This separation additionally solves the second issue mentioned above: the depth of the neural network is now fully independent of the propagation algorithm. As we saw when connecting GCN to PageRank, personalized PageRank can effectively use even infinitely many neighborhood aggregation layers, which is clearly not possible in the classical message passing framework. Furthermore, the separation gives us the flexibility to use any method for generating predictions, e.g. deep convolutional neural networks for graphs of images.

While generating predictions and propagating them happen consecutively during inference, it is important to note that **the model is trained end-to-end**. That is, the gradient flows through the propagation scheme during backpropagation (implicitly considering infinitely many neighborhood aggregation layers). Adding these propagation effects significantly improves the model's accuracy.

**Efficiency analysis.** Directly calculating the fully personalized PageRank matrix $\boldsymbol{\Pi}_{\text{ppr}}$, is computationally inefficient and results in a dense $\mathbb{R}^{n \times n}$ matrix. Using this matrix would lead to a computational complexity and memory requirement of $\mathcal{O}(n^2)$ for training and inference.

To solve this issue, reconsider the equation $\boldsymbol{Z} = \alpha(\boldsymbol{I}_n - (1-\alpha)\hat{\tilde{\boldsymbol{A}}})^{-1}\boldsymbol{H}$. Instead of viewing this equation as a combination of a dense fully *personalized* PageRank matrix with the prediction matrix, we can also view it as a variant of *topic-sensitive* PageRank, with each class corresponding to one topic (Haveliwala, 2002). In this view every *column* of $\boldsymbol{H}$ defines an (unnormalized) distribution over nodes that acts as a teleport set. Hence, we can approximate PPNP via an approximate computation of topic-sensitive PageRank.

**Approximate personalized propagation of neural predictions (APPNP).** More precisely, APPNP achieves linear computational complexity by approximating topic-sensitive PageRank via power iteration. While PageRank's power iteration is connected to the regular random walk, the power iteration of topic-sensitive PageRank is related to a random walk with restarts. Each power iteration (random walk/propagation) step of our topic-sensitive PageRank variant is, thus, calculated via

$$
\begin{aligned}
\boldsymbol{Z}^{(0)} &= \boldsymbol{H} = f_\theta(\boldsymbol{X}), \\
\boldsymbol{Z}^{(k+1)} &= (1-\alpha)\hat{\tilde{\boldsymbol{A}}}\boldsymbol{Z}^{(k)} + \alpha\boldsymbol{H}, \\
\boldsymbol{Z}^{(K)} &= \text{softmax}\left((1-\alpha)\hat{\tilde{\boldsymbol{A}}}\boldsymbol{Z}^{(K-1)} + \alpha\boldsymbol{H}\right),
\end{aligned}
\tag{4}
$$

where the prediction matrix $\boldsymbol{H}$ acts as both the starting vector and the teleport set, $K$ defines the number of power iteration steps and $k \in [0, K-2]$. Note that this method retains the graph's sparsity and never constructs an $\mathbb{R}^{n \times n}$ matrix. The convergence of this iterative scheme can be shown by investigating the resulting series (see Appendix B).

Note that the propagation scheme of this model does not require any additional parameters to train – as opposed to models like GCN, which typically require more parameters for each additional propagation layer. We can therefore propagate very far with very few parameters. Our experiments show that this ability is indeed very beneficial (see Section 6). A similar model expressed in the message passing framework would therefore not be able to achieve the same level of performance.

The reformulation of PPNP via fixed-point iterations illustrates a connection to the original graph neural network (GNN) model (Scarselli et al., 2009). While the latter uses a learned fixed-point iteration, our approach uses a predetermined iteration (adapted personalized PageRank) and applies a learned feature transformation *before* propagation.

In both PPNP and APPNP, the size of the neighborhood influencing each node can be adjusted via the teleport probability $\alpha$. The freedom to choose $\alpha$ allows us to adjust the model for different types of networks, since varying graph types require the consideration of different neighborhood sizes, as shown in Section 6 and described by Grover & Leskovec (2016) and Abu-El-Haija et al. (2018b).

Table 1: Dataset statistics. Shortest path length is denoted by SP.

| Dataset | Type | Classes | Features | Nodes | Edges | Label rate | Avg. SP |
|---|---|---|---|---|---|---|---|
| CITESEER | Citation | 6 | 3703 | 2110 | 3668 | 0.036 | 9.31 |
| CORA-ML | Citation | 7 | 2879 | 2810 | 7981 | 0.047 | 5.27 |
| PUBMED | Citation | 3 | 500 | 19 717 | 44 324 | 0.003 | 6.34 |
| MS ACADEMIC | Co-author | 15 | 6805 | 18 333 | 81 894 | 0.016 | 5.43 |

## 4 RELATED WORK

Several works have tried to improve the training of message passing algorithms and increase the neighborhood available at each node by adding skip connections (Li et al., 2016; Pham et al., 2017; Hamilton et al., 2017; Ying et al., 2018). One recent approach combined skip connection with aggregation schemes (Xu et al., 2018). However, the range of these models is still limited, as apparent in the low number of message passing layers used. While it is possible to add skip connections in the neural network used by our algorithm, this would not influence the propagation scheme. Our approach to solving the range problem is therefore unrelated to these models.

Li et al. (2018) facilitated training by combining message passing with co- and self-training. The improvements achieved by this combination are similar to results reported with other semi-supervised classification models (Buchnik & Cohen, 2018). Note that most algorithms, including ours, can be improved using self- and co-training. However, each additional step used by these methods corresponds to a full training cycle and therefore significantly increases the training time.

Deep GNNs that avoid the oversmoothing issue have been proposed in recent works by combining residual (skip) connections with batch normalization (Kawamoto et al., 2018; Chen et al., 2019). However, our model solves this issue by simplifying the architecture via decoupling prediction and propagation and does not rely on ad-hoc techniques that further complicate the model and introduce additional hyperparameters. Furthermore, since PPNP increases the range without introducing additional layers and parameters it is easier and faster to train compared to a deep GNN.

## 5 EXPERIMENTAL SETUP

Recently, many experimental evaluations have suffered from superficial statistical evaluation and experimental bias from using varying training setups and overfitting. The latter is caused by experiments using a single training/validation/test split, by not distinguishing clearly between the validation and test set, and by finetuning hyperparameters to each dataset or even data split separately. Message-passing algorithms are very sensitive to both data splits and weight initialization (as clearly shown by our evaluation). Thus, a carefully designed evaluation protocol is extremely important. Our work aims to establish such a thorough evaluation protocol. First, we run each experiment 100 times on multiple random splits and initializations. Second, we split the data into a visible and a test set, which do not change. The test set was only used *once* to report the final performance; and in particular, has never been used to perform hyperparameter and model selection. To further prevent overfitting we use the *same* number of layers and hidden units, dropout rate $d$, $L_2$ regularization parameter $\lambda$, and learning rate $l$ across datasets, since all datasets use bag-of-words as features. To prevent experimental bias we optimized the hyperparameters of *all* models individually using a grid search on CITESEER and CORA-ML and use the *same* early stopping criterion across models.

Finally, to ensure the statistical robustness of our experimental setup, we calculate confidence intervals via bootstrapping and report the p-values of a paired $t$-test for our main claims. To our knowledge, this is the most rigorous study on GCN-like models which has been done so far. More details about the experimental setup are provided in Appendix C.

**Datasets.** We use four text-classification datasets for evaluation. CITESEER (Sen et al., 2008), CORA-ML (McCallum et al., 2000; Bojchevski & Günnemann, 2018) and PUBMED (Namata et al., 2012) are citation graphs, where each node represents a paper and the edges represent citations between them. In the MICROSOFT ACADEMIC graph (Shchur et al., 2018) edges represent co-authorship. We use the largest connected component of each graph. All graphs use a bag-of-words representation of the papers' abstracts as features. While large graphs do not necessarily have a

Table 2: Average accuracy with uncertainties showing the 95 % confidence level calculated by bootstrapping. Previously reported improvements vanish on our rigorous experimental setup, while PPNP and APPNP significantly outperform the compared models on all datasets.

| Model | CITESEER | CORA-ML | PUBMED | MS ACADEMIC |
|---|---|---|---|---|
| V. GCN | $73.51 \pm 0.48$ | $82.30 \pm 0.34$ | $77.65 \pm 0.40$ | $91.65 \pm 0.09$ |
| GCN | $75.40 \pm 0.30$ | $83.41 \pm 0.39$ | $78.68 \pm 0.38$ | $92.10 \pm 0.08$ |
| N-GCN | $74.25 \pm 0.40$ | $82.25 \pm 0.30$ | $77.43 \pm 0.42$ | $92.86 \pm 0.11$ |
| GAT | $75.39 \pm 0.27$ | $84.37 \pm 0.24$ | $77.76 \pm 0.44$ | $91.22 \pm 0.07$ |
| JK | $73.03 \pm 0.47$ | $82.69 \pm 0.35$ | $77.88 \pm 0.38$ | $91.71 \pm 0.10$ |
| Bt. FP | $73.55 \pm 0.57$ | $80.84 \pm 0.97$ | $72.94 \pm 1.00$ | $91.61 \pm 0.24$ |
| PPNP* | $\mathbf{75.83 \pm 0.27}$ | $\mathbf{85.29 \pm 0.25}$ | - | - |
| APPNP | $75.73 \pm 0.30$ | $85.09 \pm 0.25$ | $\mathbf{79.73 \pm 0.31}$ | $\mathbf{93.27 \pm 0.08}$ |

*out of memory on PUBMED, MS ACADEMIC (see efficiency analysis in Section 3)

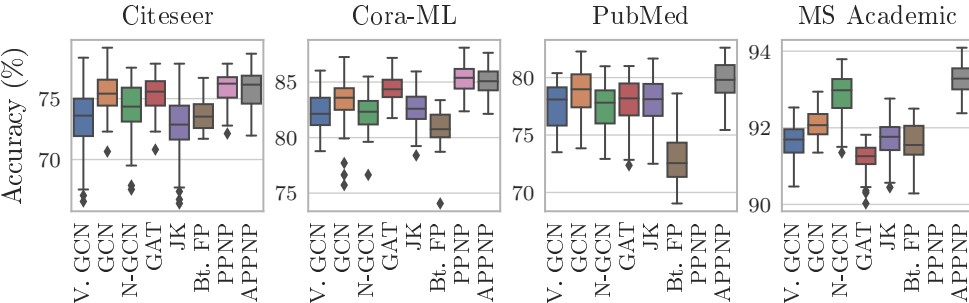

Figure 2: Accuracy distributions of different models. The high standard deviation between data splits and initializations shows the importance of a rigorous evaluation, which is often omitted.

larger diameter (Leskovec et al., 2005), note that these graphs indeed have average shortest path lengths between 5 and 10 and therefore a regular two-layer GCN cannot cover the entire graph. Table 1 reports the dataset statistics.

**Baseline models.** We compare to five state-of-the-art models: GCN (Kipf & Welling, 2017), network of GCNs (N-GCN) (Abu-El-Haija et al., 2018a), graph attention networks (GAT) (Veličković et al., 2018), bootstrapped feature propagation (bt. FP) (Buchnik & Cohen, 2018) and jumping knowledge networks with concatenation (JK) (Xu et al., 2018). For GCN we also show the results of the (unoptimized) vanilla version (V. GCN) to demonstrate the strong impact of early stopping and hyperparameter optimization. The hyperparameters of all models are listed in Appendix D.

**Model hyperparameters.** To ensure a fair model comparison we used a neural network for PPNP that is structurally very similar to GCN and has the same number of parameters. We use two layers with $h = 64$ hidden units. We apply $L_2$ regularization with $\lambda = 0.005$ on the weights of the first layer and use dropout with dropout rate $d = 0.5$ on both layers and the adjacency matrix. For APPNP, adjacency dropout is resampled for each power iteration step. For propagation we use the teleport probability $\alpha = 0.1$ and $K = 10$ power iteration steps for APPNP. We use $\alpha = 0.2$ on the MICROSOFT ACADEMIC graph due to its structural difference (see Figure 5 and its discussion). The combination of this shallow neural network with a comparatively high number of power iteration steps achieved the best results during hyperparameter optimization (see Appendix G).

## 6 RESULTS

**Overall accuracy.** The results for the accuracy (micro F1-score) are summarized in Table 2. Similar trends are observed for the macro F1-score (see Appendix E). Both models significantly outperform the state-of-the-art baseline models on all datasets. Our rigorous setup might understate the improvements achieved by PPNP and APPNP – this result is statistically significant $p < 0.05$, as tested via a paired $t$-test (see Appendix F). This thorough setup furthermore shows that the advantages reported by recent works practically vanish when training is harmonized, hyperparameters are properly op-

Table 3: Average training time per epoch. PPNP and APPNP are only slightly slower than GCN and much faster than more sophisticated methods like GAT.

| Graph | V. GCN | GCN | N-GCN | GAT | JK | Bt. FP* | PPNP** | APPNP |
|---|---|---|---|---|---|---|---|---|
| CITESEER | 37.6 ms | 35.3 ms | 115.9 ms | 187.0 ms | 57.5 ms | - | 49.2 ms | 43.3 ms |
| CORA-ML | 32.4 ms | 36.5 ms | 118.9 ms | 217.4 ms | 43.6 ms | - | 55.3 ms | 42.7 ms |
| PUBMED | 48.6 ms | 48.3 ms | 342.6 ms | 1029.8 ms | 77.8 ms | - | - | 64.1 ms |
| MS ACADEMIC | 45.5 ms | 39.2 ms | 328.5 ms | 772.2 ms | 61.9 ms | - | - | 59.8 ms |

*not applicable, since core method not trainable    **out of memory on PUBMED, MS ACADEMIC (see efficiency analysis in Section 3)

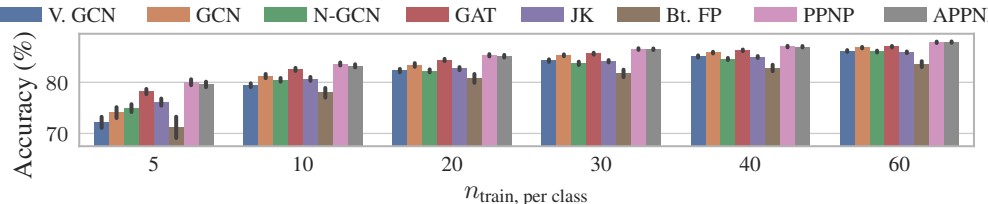

Figure 3: Accuracy for different training set sizes (number of labeled nodes per class) on CORA-ML. PPNP's dominance increases further for smaller training set sizes.

timized and multiple data splits are considered. A simple GCN with optimized hyperparameters outperforms several recently proposed models on our setup.

Figure 2 shows how broad the accuracy distribution of each model is. This is caused by both random initialization and different data splits (train / early stopping / test). This demonstrates how crucial a statistically rigorous evaluation is for a conclusive model comparison. Moreover, it shows the sensitivity (robustness) of each method, e.g. PPNP, APPNP and GAT typically have lower variance.

**Training time per epoch.** We report the average training time per epoch in Table 3. We decided to only compare the training time per epoch since all hyperparameters were solely optimized for accuracy and the used early stopping criterion is very generous. Obviously, (exact) PPNP can only be applied to moderately sized graphs, while APPNP scales to large data. On average, APPNP is around 25 % slower than GCN due to its higher number of matrix multiplications. It scales similarly with graph size as GCN and is therefore significantly faster than other more sophisticated models like GAT. This is observed even though our implementation improved GAT's training time roughly by a factor of 2 compared to the reference implementation.

**Training set size.** Since the labeling rate is often very small for real world datasets, investigating how the models perform with a small number of training samples is very important. Figure 3 shows how the number of training nodes per class $n_{\text{train, per class}}$ impacts the accuracy on CORA-ML (for other datasets see Appendix H). The dominance of PPNP and APPNP increases further in this sparsely labelled setting. This can be attributed to their higher range, which allows them to better propagate the information further away from the (few) training nodes. We see further evidence for this when comparing the accuracy of APPNP and GCN depending on the distance between a node and the training set (in terms of shortest path). Appendix I shows that the performance gap between APPNP and GCN tends to increase for nodes that are far away from the training nodes. That is, nodes further away from the training set benefit more from the increase in range.

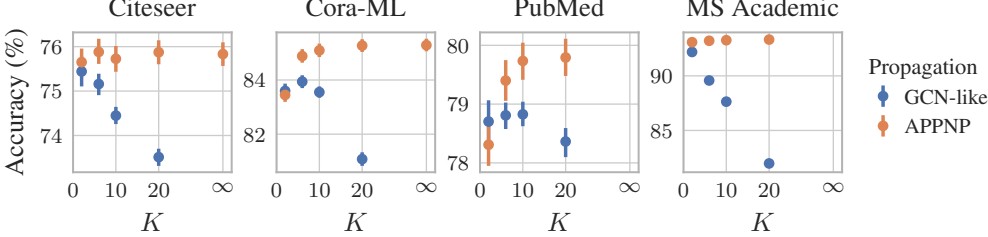

Figure 4: Accuracy depending on the number of propagation steps $K$. The accuracy breaks down for the GCN-like propagation ($\alpha = 0$), while it increases and stabilizes when using APPNP ($\alpha = 0.1$).

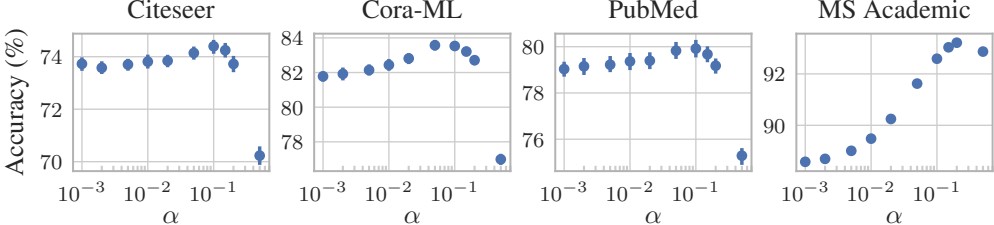

Figure 5: Accuracy depending on teleport probability $\alpha$. The optimum typically lies within $\alpha \in [0.05, 0.2]$, but changes for different types of datasets.

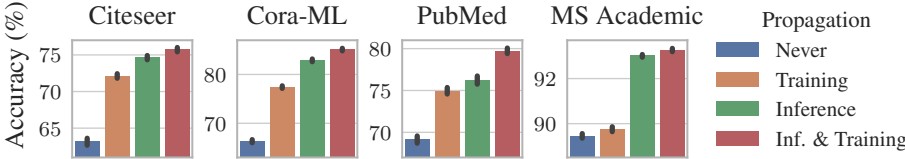

Figure 6: Accuracy of APPNP with propagation used only during training/inference. Best results are achieved with full propagation, but propagating only during inference also achieves good results.

**Number of power iteration steps.** Figure 4 shows how the accuracy depends on the number of power iterations for two different propagation schemes. The first mimics the standard propagation as known from GCNs (i.e. $\alpha = 0$ in APPNP). As clearly shown the performance breaks down as we increase the number of power iterations $K$ (since we approach the global PageRank solution). However, when using personalized propagation (with $\alpha = 0.1$) the accuracy *increases* and converges to exact PPNP with infinitely many propagation steps, thus demonstrating the personalized propagation principle is indeed beneficial. As also shown in the figure, it is enough to use a moderate number of power iterations (e.g. $K = 10$) to effectively approximate exact PPNP. Interestingly, we've found that this number coincides with the highest shortest path distance of any node to the training set.

**Teleport probability $\alpha$.** Figure 5 shows the effect of the hyperparameter $\alpha$ on the accuracy on the validation set. While the optimum differs slightly for every dataset, we consistently found a teleport probability of around $\alpha \in [0.05, 0.2]$ to perform best. This probability should be adjusted for the dataset under investigation, since different graphs exhibit different neighborhood structures (Grover & Leskovec, 2016; Abu-El-Haija et al., 2018b). Note that a higher $\alpha$ improves convergence speed.

**Neural network without propagation.** PPNP and APPNP are trained end-to-end, with the propagation scheme affecting (i) the neural network $f_\theta$ during training, and (ii) the classification decision during inference. Investigating how the model performs without propagation shows if and how valuable this addition is. Figure 6 shows how propagation affects both training and inference. "Never" denotes the case where no propagation is used; essentially we train and apply a standard multilayer perceptron (MLP) $f_\theta$ using the features only. "Training" denotes the case where we use APPNP during training to learn $f_\theta$; at inference time, however, only $f_\theta$ is used to predict the class labels. "Inference", in contrast, denotes the case where $f_\theta$ is trained without APPNP (i.e. standard MLP on features). This pretrained network with fixed weights is then used with APPNP's propagation for inference. Finally, "Inf. & Training" denotes the regular APPNP, which always uses propagation.

The best results are achieved with regular APPNP, which validates our approach. However, on most datasets the accuracy decreases surprisingly little when propagating only during inference. Skipping propagation during training can significantly reduce training time for large graphs as all nodes can be handled independently. This also shows that our model can be combined with pretrained neural networks that do not incorporate any graph information and still significantly improve their accuracy. Moreover, Figure 6 shows that just propagating during training can also lead to large improvements. This indicates that our model can also be applied to online/inductive learning where only the features and not the neighborhood information of an incoming (previously unobserved) node are available.

## 7 CONCLUSION

In this paper we have introduced personalized propagation of neural predictions (PPNP) and its fast approximation, APPNP. We derived this model by considering the relationship between GCN and PageRank and extending it to personalized PageRank. This simple model decouples prediction and propagation and solves the limited range problem inherent in many message passing models without introducing any additional parameters. It uses the information from a large, adjustable (via the teleport probability $\alpha$) neighborhood for classifying each node. The model is computationally efficient and outperforms several state-of-the-art methods for semi-supervised classification on multiple graphs in the most thorough study which has been done for GCN-like models so far.

For future work it would be interesting to combine PPNP with more complex neural networks used e.g. in computer vision or natural language processing. Furthermore, faster or incremental approximations of personalized PageRank (Bahmani et al., 2010; 2011; Lofgren et al., 2014) and more sophisticated propagation schemes would also benefit the method.

## ACKNOWLEDGEMENTS

This research was supported by the German Research Foundation, grant GU 1409/2-1.

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

## A  EXISTENCE OF $\boldsymbol{\Pi}_{\text{PPR}}$

The matrix

$$\boldsymbol{\Pi}_{\text{ppr}} = \alpha \left( \boldsymbol{I}_n - (1-\alpha)\hat{\tilde{\boldsymbol{A}}} \right)^{-1} \tag{5}$$

exists iff the determinant $\det(\boldsymbol{I}_n - (1-\alpha)\hat{\tilde{\boldsymbol{A}}}) \neq 0$, which is the case iff $\det(\hat{\tilde{\boldsymbol{A}}} - \frac{1}{1-\alpha}\boldsymbol{I}_n) \neq 0$, i.e. iff $\frac{1}{1-\alpha}$ is not an eigenvalue of $\hat{\tilde{\boldsymbol{A}}}$. This value is always larger than 1 since the teleport probability $\alpha \in (0, 1]$. Furthermore, the symmetrically normalized matrix $\hat{\tilde{\boldsymbol{A}}}$ has the same eigenvalues as the row-stochastic matrix $\tilde{\boldsymbol{A}}_{\text{rw}}$. This can be shown by multiplying the eigenvalue equation $\hat{\tilde{\boldsymbol{A}}}\boldsymbol{v} = \lambda\boldsymbol{v}$ with $\tilde{\boldsymbol{D}}^{-1/2}$ from left and substituting $\boldsymbol{w} = \tilde{\boldsymbol{D}}^{-1/2}\boldsymbol{v}$. This also shows that the eigenvectors of $\hat{\tilde{\boldsymbol{A}}}$ are the eigenvectors of $\tilde{\boldsymbol{A}}_{\text{rw}}$ scaled by $\tilde{\boldsymbol{D}}^{1/2}$. The largest eigenvalue of a row-stochastic matrix is 1, as can be proven using the Gershgorin circle theorem. Hence, $\frac{1}{1-\alpha}$ cannot be an eigenvalue and $\boldsymbol{\Pi}_{\text{ppr}}$ always exists.

## B    CONVERGENCE OF APPNP

APPNP uses the iterative equation

$$\boldsymbol{Z}^{(k+1)} = (1-\alpha)\hat{\tilde{\boldsymbol{A}}}\boldsymbol{Z}^{(k)} + \alpha\boldsymbol{H}. \tag{6}$$

After the $k$-th propagation step, the resulting predictions are

$$\boldsymbol{Z}^{(k)} = \left( (1-\alpha)^k \hat{\tilde{\boldsymbol{A}}}^k + \alpha \sum_{i=0}^{k-1} (1-\alpha)^i \hat{\tilde{\boldsymbol{A}}}^i \right) \boldsymbol{H}. \tag{7}$$

If we take the limit $k \to \infty$ the left term tends to 0 and the right term becomes a geometric series. The series converges since $\alpha \in (0, 1]$ and $\hat{\tilde{\boldsymbol{A}}}$ is symmetrically normalized and therefore $\det(\hat{\tilde{\boldsymbol{A}}}) \leq 1$, resulting in

$$\boldsymbol{Z}^{(\infty)} = \alpha \left( \boldsymbol{I}_n - (1-\alpha)\hat{\tilde{\boldsymbol{A}}} \right)^{-1} \boldsymbol{H}, \tag{8}$$

which is the equation for calculating (exact) PPNP.

## C    EXPERIMENTAL DETAILS

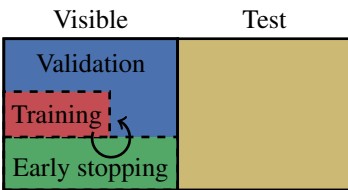

Figure 7: Illustration of the node sampling procedure.

The sampling procedure is illustrated in Figure 7. The data is first split into a visible and a test set. For the visible set 1500 nodes were sampled for the citation graphs and 5000 for MICROSOFT ACADEMIC. The test set contains all remaining nodes. We use three different label sets in each experiment: A training set of 20 nodes per class, an early stopping set of 500 nodes and either a validation or test set. The validation set contains the remaining nodes of the visible set. We use 20 random seeds for determining the splits. These seeds are drawn once and fixed across runs to facilitate comparisons. We use one set of seeds for the validation splits and a different set for the test splits. Each experiment is run with 5 random initializations on each data split, leading to a total of 100 runs per experiment.

The early stopping criterion uses a patience of $p = 100$ and an (unreachably high) maximum of $n = 10\,000$ epochs. The patience is reset whenever the accuracy increases or the loss decreases on the early stopping set. We choose the parameter set achieving the highest accuracy and break ties by selecting the lowest loss on this set. This criterion was inspired by GAT (Veličković et al., 2018).

We used TensorFlow (Martín Abadi et al., 2015) for all experiments except bootstrapped feature propagation. All uncertainties and confidence intervals correspond to a confidence level of 95 % and were calculated by bootstrapping with 1000 samples.

We use the Adam optimizer with a learning rate of $l = 0.01$ and cross-entropy loss for all models (Kingma & Ba, 2015). Weights are initialized as described in Glorot & Bengio (2010). The feature matrix is $L_1$ normalized per row.

## D  BASELINE HYPERPARAMETERS

Vanilla GCN uses the original settings of two layers with $h = 16$ hidden units, no dropout on the adjacency matrix, $L_2$ regularization parameter $\lambda = 5 \times 10^{-4}$ and the original early stopping with a maximum of 200 steps and a patience of 10 steps based on the loss.

The optimized GCN uses two layers with $h = 64$ hidden units, dropout on the adjacency matrix with $d = 0.5$ and $L_2$ regularization parameter $\lambda = 0.02$.

N-GCN uses $h = 16$ hidden units, $R = 4$ heads per random walk length and random walks of up to $K - 1 = 4$ steps. It uses $L_2$ regularization on all layers with $\lambda = 1 \times 10^{-5}$ and the attention variant for merging the predictions (Abu-El-Haija et al., 2018a). Note that this model effectively uses $RKh = 320$ hidden units, which is 5 times as many units compared to GCN, GAT, and PPNP.

For GAT we use the (well optimized) original hyperparameters, except the $L_2$ regularization parameter $\lambda = 0.001$ and learning rate $l = 0.01$. As opposed to the original paper, we do not use different hyperparameters on PUBMED, as described in our experimental setup.

Bootstrapped feature propagation uses a return probability of $\alpha = 0.2$, 10 propagation steps, 10 bootstrapping (self-training) steps with $r = 0.1n$ training nodes added per step. We add the training nodes with the lowest entropy on the predictions. The number of nodes added per class is based on the class proportions estimated using the predictions. Note that this model does not include any stochasticity in its initialization. We therefore only run it once per train/early stopping/test split.

For the jumping knowledge networks we use the concatenation variant with three layers and $h = 64$ hidden units per layer. We apply $L_2$ regularization with $\lambda = 0.001$ on all layers and perform dropout with $d = 0.5$ on all layers but not on the adjacency matrix.

## E  F1 SCORE

Table 4: Average macro F1 score with uncertainties showing the $95\,\%$ confidence level calculated by bootstrapping. PPNP achieves the highest F1 score on all datasets investigated.

| Model | CITESEER | CORA-ML | PUBMED | MS ACADEMIC |
|---|---|---|---|---|
| V. GCN | $0.7002 \pm 0.0043$ | $0.8205 \pm 0.0027$ | $0.7801 \pm 0.0038$ | $0.9000 \pm 0.0008$ |
| GCN | $0.7065 \pm 0.0037$ | $0.8289 \pm 0.0030$ | $0.7883 \pm 0.0032$ | $0.9045 \pm 0.0008$ |
| N-GCN | $0.7021 \pm 0.0035$ | $0.8183 \pm 0.0024$ | $0.7773 \pm 0.0040$ | $0.9144 \pm 0.0012$ |
| GAT | $0.7062 \pm 0.0029$ | $0.8359 \pm 0.0025$ | $0.7777 \pm 0.0040$ | $0.8917 \pm 0.0007$ |
| JK | $0.6914 \pm 0.0043$ | $0.8202 \pm 0.0026$ | $0.7799 \pm 0.0039$ | $0.8985 \pm 0.0012$ |
| Bt. FP | $0.6789 \pm 0.0055$ | $0.8026 \pm 0.0082$ | $0.7448 \pm 0.0079$ | $0.8997 \pm 0.0018$ |
| PPNP* | $0.7102 \pm 0.0041$ | $\mathbf{0.8454 \pm 0.0021}$ | - | - |
| APPNP | $\mathbf{0.7105 \pm 0.0038}$ | $0.8429 \pm 0.0022$ | $\mathbf{0.7966 \pm 0.0031}$ | $\mathbf{0.9184 \pm 0.0009}$ |

$^*$out of memory on PUBMED, MS ACADEMIC (see efficiency analysis in Section 3)

## F  PAIRED $t$-TEST

Table 5: p-value of the paired $t$-test with respect to accuracy.

| Model | CITESEER | CORA-ML | PUBMED | MS ACADEMIC |
|---|---|---|---|---|
| PPNP | $1.02 \times 10^{-3}$ | $5.96 \times 10^{-14}$ | - | - |
| APPNP | $1.77 \times 10^{-2}$ | $4.27 \times 10^{-9}$ | $2.19 \times 10^{-15}$ | $5.93 \times 10^{-13}$ |

Table 6: p-value of the paired $t$-test with respect to F1 score.

| Model | CITESEER | CORA-ML | PUBMED | MS ACADEMIC |
|---|---|---|---|---|
| PPNP | $4.49 \times 10^{-2}$ | $4.50 \times 10^{-14}$ | - | - |
| APPNP | $2.32 \times 10^{-2}$ | $1.07 \times 10^{-8}$ | $8.70 \times 10^{-14}$ | $1.99 \times 10^{-8}$ |

# G    NUMBER OF NEURAL NETWORK LAYERS

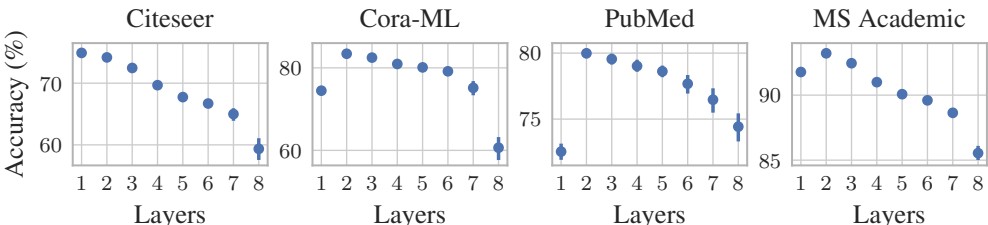

Figure 8: Validation accuracy of APPNP for varying numbers of neural network (NN) layers. Deep NNs do not improve the accuracy, which is probably due to the simple bag-of-words features and the small training set size.

# H    TRAINING SET SIZE

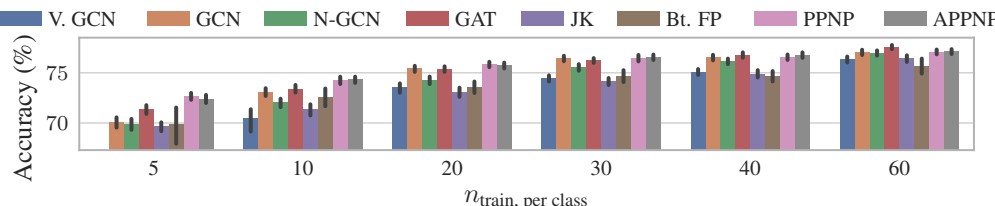

Figure 9: Accuracy for different training set sizes on CITESEER.

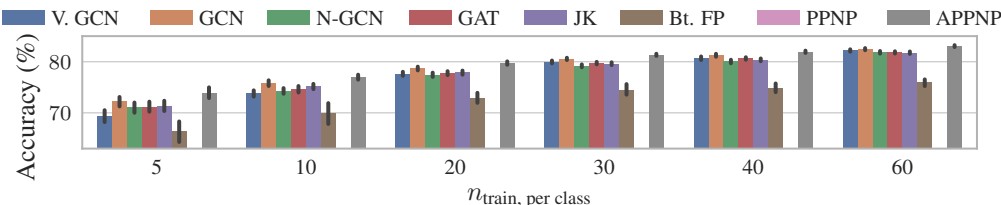

Figure 10: Accuracy for different training set sizes on PUBMED.

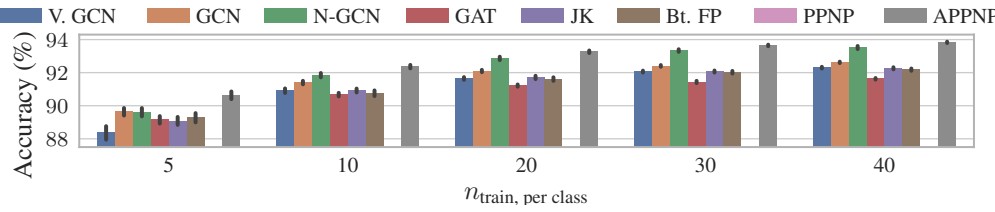

Figure 11: Accuracy for different training set sizes on MICROSOFT ACADEMIC.

# I   ACCURACY DEPENDING ON DISTANCE FROM TRAINING NODES

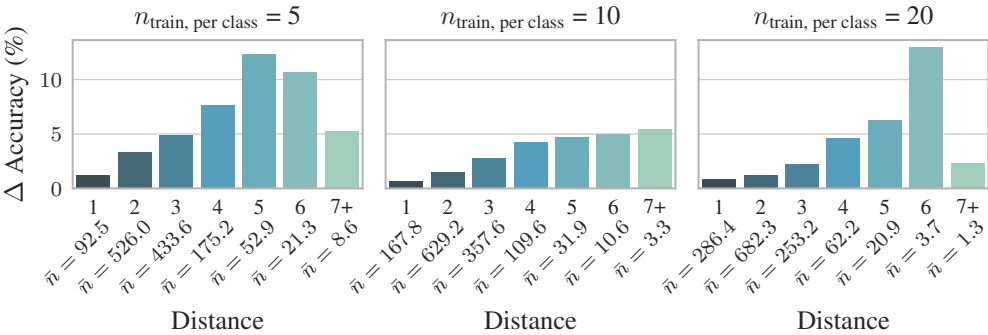

Figure 12: $\Delta$ Accuracy (%) denotes the average improvement in percentage points of APPNP over GCN depending on the distance (number of hops) from the training nodes on CORA-ML. $\bar{n}$ denotes the average number of nodes at each distance. The improvement increases with distance.

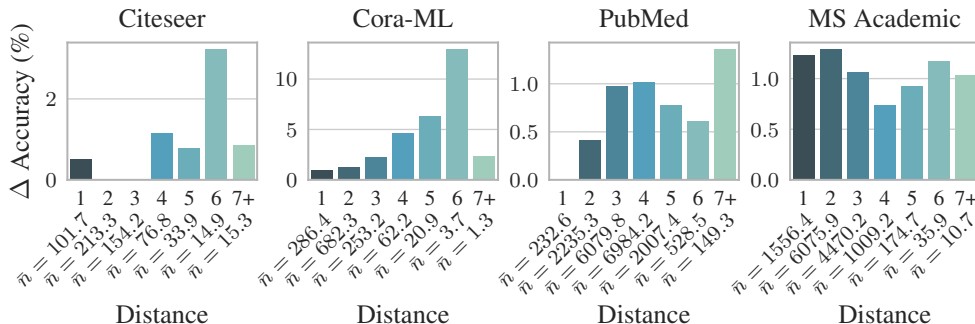

Figure 13: $\Delta$ Accuracy (%) denotes the average improvement in percentage points of APPNP over GCN depending on the distance (number of hops) from the training nodes on different graphs. $\bar{n}$ denotes the average number of nodes at each distance over different splits.

