# OpenReview forum: "Predict then Propagate: Graph Neural Networks meet Personalized PageRank"
_ICLR.cc/2019/Conference_

### Official Review · AnonReviewer3 · 2018-10-31
**Idea is interesting; experiments are convincing**

**Rating:** 7
**Confidence:** 4

**Review:**

This paper proposes a GCN variant that addresses a limitation of the original model, where embedding is propagated in only a few hops. The architectural difference may be explained in the following: GCN interleaves the individual node feature transformation and the single-hop propagation, whereas the proposed architecture first transforms the node features, followed by a propagation with an (in)finite number of hops. The propagation in the proposed method follows personalized PageRank, where in addition to following direct links, there is a nonzero probably jumping to a target node.

I find the idea interesting. The experiments are comprehensive, covering important points including data split, training set size, number of hops, teleport probability, and ablation study. Two interesting take-home messages are that (1) GCN-like propagation without teleportation leads to degrading performance as the number of hops increases, whereas propagation with teleportation leads to converging performance; and (2) the best-performing teleport probability generally falls within a narrow range.

Question: The current propagation approach uses the normalized adjacency matrix proposed by GCN, which is, strictly speaking, not the transition matrix used by PageRank. What prevents from using the transition matrix? Note that this matrix naturally handles directed graphs.

---

> ### Author Response · Authors · 2018-11-13
> **Re: Reviewer3**
>
> Thank you for your review and feedback!
>
> You are right, nothing prevents the model from using the standard transition matrix. During model development, however, we have found that the added self-loops of the GCN-matrix are beneficial to performance. The symmetrical normalization actually doesn't make any difference in the limit k->infinity. However, we found this style of normalization to be beneficial for the finite-step approximation.

---

### Official Review · AnonReviewer2 · 2018-11-02
**review on "Personalized Embedding Propagation: Combining Neural Networks on Graphs with Personalized PageRank"**

**Rating:** 5
**Confidence:** 4

**Review:**

This paper proposed a variant of graph neural network, which added additional pagerank-like propagations (with constant aggregation weights), in additional to the normal message-passing like propagation layers. Experiments on some benchmark transductive node classification tasks show some empirical gains.

Using more propagations with constant aggregation weights is an interesting idea to help propagate the information in a graph. However, this idea is not completely new. In the very first graph neural network [1], the propagation is done until convergence. If the operator in each layer is a contraction map, then according to the Banach Fixed Point theorem [2], a unique solution can be guaranteed. The constant operator used in this paper is thus a special case of this contraction map.

Also, the closed form solution in (3) is not practical. It may not be suitable for large graphs (e.g., graphs with >10k nodes). And that’s why this approach is not suitable for Pubmed and Microsoft dataset. The PEP_A is more practical. However, in this case I’m curious how it would compare with a GNN having same number of layers, but with proper gating/skip connections like ResNet.

The experiments show some marginal gains on the small graphs. However, I think it would be important to test on large graphs. Since small graphs typically have small diameter, thus several GNN layers would already cover the entire graph, and the additional propagation done by pagerank here might not be super helpful.

Finally, I think the author should properly cite another relevant paper [3], which uses fixed point iteration to help propagate the local information.

[1] Scarselli et.al, “The Graph Neural Network Model”, IEEE Transactions on Neural Networks, 2009
[2] Mohamed A. Khamsi, An Introduction to Metric Spaces and Fixed Point Theory
[3] Dai et.al, Learning Steady-States of Iterative Algorithms over Graphs, ICML 2018

---

> ### Author Response · Authors · 2018-11-13
> **Re: Reviewer2**
>
> Thank you for your review and feedback!
>
> The connection to the GNN-framework is certainly interesting and we’ve added it in the revised version of the paper (in Section 3, after introducing APPNP). However, our main contribution is not the usage of fixed-point iterations for node classification, which has already been used e.g. in label propagation and belief propagation algorithms. Our contribution is the improvement of GCN-like models by solving the limited range problem through the development and thorough evaluation of an end-to-end trained model utilizing one specific fixed-point iteration.
>
> As you correctly noticed, the exact model is not applicable to larger data -- this is exactly the reason why we have developed its approximation. The discussion can be found under "efficiency analysis" in Section 3. We have edited the experimental section to make this more clear. Furthermore, we would like to highlight that we have already performed an analysis on large graphs. As shown in Table 1, our experimental evaluation includes two graphs with 20k nodes, which follows the suggestion you gave (>10k nodes).
>
> Please note that we have already compared our model to jumping knowledge networks (JK), which is similar to the GNN that uses proper gating/skip connections you suggested. As we show in the experimental section, we significantly outperform this model.
>
> You state that we show "some marginal gains". However, we show that our results are significant. Previous methods have reported “large” gains that actually were not statistically significant and vanish when thoroughly evaluated, as we show in the paper. We paid a lot of attention to performing a fair comparison and a rigorous statistical analysis of our results, which shows that we significantly outperform previous models. The different evaluation may make the improvements seem smaller. But in fact they are larger than those reported in previous, less careful evaluations. We have edited the section to further clarify this. Furthermore, we’ve included a reference to the work by Dai et al.

---

> > ### Comment · AnonReviewer2 · 2018-11-13
> > **Re: Re: Reviewer2**
> >
> > Thanks for your reply!
> >
> > To reiterate my questions:
> >
> > 1) The graph with ~10k nodes would be the limit for your exact algorithm, as the results are missing in Table 2. But since you have the approximation with power-iteration like layers, it would be better if you can target on large graphs.
> >
> > 2) And I expect your algorithm would benefit more on large graphs. This is the case where the pagerank could be more effective in propagating information, than parameterized message passing operators. So that's why it is important to do large scaled experiments to show the truly 'significant' gains.
> >
> > 3) Here are several good large datasets you may want to take a look: https://snap.stanford.edu/data/

---

> > > ### Author Response · Authors · 2018-11-15
> > > **Re: Reviewer2**
> > >
> > > Thank you for your quick response!
> > >
> > > If we understand you correctly, all your points above are referring to the study of larger graphs to ensure a large diameter (since, as mentioned in your first comment, a large diameter requires more propagation steps). Note, however, that the graph diameter usually shrinks with graph size (see e.g. Leskovec 2005). Thus, instead of studying even larger graphs one should analyze graphs with sufficiently large diameter. Indeed, the graphs we have already studied in our paper have an average diameter between 5 and 10 (see Table 1 of the revised version). Thus, a few GCN layers can not cover the entire graph.
> > >
> > > Our experiments further show that denser graphs with a smaller diameter (e.g. Microsoft Academic) require a higher alpha (see Figure 5). Your discussion actually prompted us to adjust alpha on this dataset to better reflect the graph’s underlying characteristics (see Section 6 of the revised version).
> > >
> > > Furthermore, we are not sure what exactly you mean with ‘significant’ -- and why you have the impression that our results are not significant. In our paper and comments we use the term significant in the mathematical sense of statistical significance. The results clearly show that our method’s improvements are significant with a p-value of 0.05, as we have shown in our rigorous evaluation (for small and large graphs as well as graphs with different diameters).

---

> > > > ### Comment · Area_Chair1 · 2018-12-08
> > > > **clarification**
> > > >
> > > > I believe the reviewer here meant "substantial and practically meaningful" and not "statistically significant."
> > > >
> > > > Your point about graph diameter is a good one. However I am wondering if you can elaborate a bit on your argument in section 2 where you say:
> > > >
> > > > "There are essentially two reasons why a message passing algorithm like GCN can’t be trivially expanded to use a larger neighborhood. First, aggregation by averaging causes oversmoothing if too many layers are used. It, therefore, loses its focus on the local neighborhood (Li et al., 2018). Second, most common aggregation schemes use learnable weight matrices in each layer. Therefore, using a larger neighborhood necessarily increases the depth and number of learnable parameters of the neural network (the second aspect can be circumvented by using weight sharing, which is typically not the case, though)."
> > > >
> > > > It seems fine to use weight sharing to deal with the second issue and I believe it isn't that uncommon. However, the oversmoothing issue could be a larger problem. Couldn't this be dealt with using attention-like mechanisms or different aggregation functions like max instead of sum (or intermediate functions)?
> > > >
> > > > An average diameter of 10, the largest for datasets you explore, might not be enough to be problematic. Keeping in mind that I have not carefully read the paper, only skimmed it, can you succinctly summarize what evidence you have that limited range is an important issue in practice? I agree with the premise that it could be (because of the tying network depth or recurrent sequence length to neighborhood size is somewhat arbitrary), but I am wondering how best to demonstrate this is an issue and your approach is a successful solution on an important problem of practical interest.

---

> > > > > ### Author Response · Authors · 2018-12-10
> > > > > **The issues of limited range and oversmoothing**
> > > > >
> > > > > ** Issue of Limited Range **
> > > > >
> > > > > Evidence for showing that larger neighborhoods are beneficial is shown e.g. in Figures 4 and 5 of the paper. Figure 4 shows how the accuracy increases dramatically on Cora-ML and PubMed as we increase the number of propagation steps beyond 2. Figure 5 shows that the optimal α lies between 0.05 and 0.2. For these values, between 86% and 51% of the influence comes from neighborhoods using more than 2 propagation steps.
> > > > >
> > > > > Furthermore, larger neighborhoods are especially important in the sparsely labelled setting, as shown by Li, Han and Wu (AAAI 2018) and in Figure 3 of our paper. This figure shows that our method can handle small training sets best and outperforms GCN by 6 percentage points in this setting.
> > > > >
> > > > > Xu et al. (ICML 2018) have also found the limited range to be an issue, especially for nodes in the periphery. Very little information will reach these nodes with only 2 hops and a higher range is therefore critical for classifying these.
> > > > >
> > > > > ** Oversmoothing and attention-like mechanisms **
> > > > >
> > > > > An attention-like mechanism for working with multiple different neighborhood sizes was already investigated in previous work by Xu et al. in the jumping knowledge networks (JK) model. However, for most experiments they still achieved best performance when using only 2-3 layers. In our own experiments we have found JK to perform best with only 3 layers and in the paper we show that our new model significantly outperforms it.
> > > > >
> > > > > In earlier experiments we have also tested attention over different neighborhood sizes in combination with our model, but found that learning the attention weights is problematic and mostly overfits on the node itself. Please note that our personalized PageRank uses an *implicit* attention scheme on the different neighborhoods with weights α(1-α)^k (for the k-step neighborhood), which we have found to perform significantly better than any other weighting scheme we have tested. This implicit attention mechanism might be one reason why our model performs so well.
> > > > >
> > > > > We have also experimented with increasing the number of layers in GAT (which uses attention for its node aggregation function), but were not able to successfully increase its number of layers beyond the original 2.
> > > > >
> > > > > Finally, different node aggregation functions were used by e.g. GraphSAGE, which also performs best when using no more than 2 layers and therefore shows the same problem of limited range.

---

### Official Review · AnonReviewer1 · 2018-11-03
**Interesting but limited contribution**

**Rating:** 5
**Confidence:** 4

**Review:**

The thurst behind this paper is that graph convolutional networks (GCNs) are constrained by construction
to focus on small neighborhoods around any given node. Large neighborhoods introduce in principle
a large number of parameters (while as the authors point out, weight sharing is an option to avoid this issue),
plus even worse oversmoothing may occur. Specifically, Xu et al. (2018) showed that for a k-layer GCN one can
think of the influence score of a node x on node y as the probability  that a walker that starts at x,
lands on y after k steps of random walk (modulo some details).

Therefore, as k increases the random walks reaches its stationary distribution, forgetting any local information that is useful,
e.g., for node classification. To avoid this problem, the authors propose the following: use personalized Pagerank
instead of the standard Markov chain of Pagerank. In PPR there is a restart probability, which allows
their algorithm to avoid “forgetting” the local information around a walk, thus allowing for an arbitrary
number of steps in their random walk. The authors define two methods PEP, and PEPa based on PPR. The latter
method is faster in practice since it approximates the PPR.

A key advantage of the proposed method is the separation of the node embedding part from the propagation scheme. In this sense,
following the categorization of existing methods into three categories, PEP is a hybrid of message passing algorithms,
and random walk based node embeddings. The experimental evaluation tests certain basic properties of the proposed method. One interesting performance feature of
PEP and PEPa is that they can perform well using few training examples. This is valuable especially when obtaining labeled
examples is expensive.  Finally, the authors compare their proposed methods against state-of-the-art GCN-based methods.

Some remarks follow.

- The idea of using PPR for node embeddings has been suggested in recent prior work “LASAGNE: Locality and structure aware graph node embeddings”
By Faerman et al.  While according to the authors’ categorization of the existing methods in the intro, LASAGNE
falls under the “random walk” family  of methods, the authors should compare against it.

- Continuing the previous point,  even simpler baselines would be desirable. How inferior is for instance
an approach on one-vs-all classification using the approximate personalized Pagerank node embedding and
support vector machines?

- Also, the authors mention “since our datasets are somewhat similar…”. Please clarify with respect to
which aspects? Also, please use datasets that are different. For instance, see the LASAGNE paper for
more datasets that have different number of classes.

- In the experiments the authors use two layers for fair comparison. Given that one of the advantages of the
proposed method is the  ability to have more layers without suffering from the GCN shortcomings
with large neighborhood exploration, it would be interesting to see an experiment where the number of layers is a variable.

---

> ### Author Response · Authors · 2018-11-13
> **Re: Reviewer1**
>
> Thank you for your review and feedback!
>
> We want to clarify that the principle and task performed by LASAGNE is fundamentally different to ours. The LASAGNE method learns individual node embeddings in an unsupervised setting. Our goal is not to learn individual node embeddings but to learn a transformation from attributes to class labels in the semi-supervised setting, as graph convolutional network (GCN)-like models do. Moreover, LASAGNE only considers structural information. Generally, it has been shown that approaches that consider both structure and attributes outperform methods that only consider the structure (see e.g. Kipf Welling 2017). Therefore, we only compare with methods that consider both, but we added a reference to LASAGNE in the paper.
>
> We feel that this confusion was due to a bad framing of our model. To make things clearer we have decided to rename the model and replace the term “embedding” with “prediction” in the revised version (see also our general comment).
>
> We cannot run the proposed baseline, since as we clarified above we do not learn any personalized pagerank embeddings to begin with. However, we do already include a comparatively simple baseline which is the bootstrapped Laplacian feature propagation. This method propagates features in a similar way as we do and then uses a one-vs-all classifier. We significantly outperform this baseline.
>
> In the revised version of the paper we clarified that the datasets are similar in that they contain bag-of-words features and use scientific networks. However, these graphs have very different numbers of nodes, edges, features, and classes, and different topology, as shown in Table 1. The datasets you suggested from the LASAGNE paper are not suitable for the kind of semi-supervised classification we consider since they do not contain node attributes.
>
> Thank you for suggesting the interesting experiment of varying neural network depth! The investigated datasets do not benefit from deeper networks. You can find the results in Figure 11 of the updated version of the paper.

---

> > ### Comment · AnonReviewer1 · 2018-12-08
> > **thanks for your response**
> >
> > Dear authors,
> >
> > I would like to thank you for the detailed response(s) to the review(s you have received). I would also like to make a related comment: I agree with your comments overall, modulo that there has been any confusion. Your experimental setup was clear from the first time I read your nice paper, that is why I mentioned in one of comments that " While according to the authors’ categorization of the existing methods in the intro, LASAGNE falls under the “random walk” family  of methods".    Perhaps I should have made it more clear in my review, that personally as a reviewer I would have liked to see some basic classification baseline that is related to PPR, that was my main point and why I made  two possible suggestions.
> >
> > I have upgraded my score. I want to clarify that my non-acceptance score as my review title summarized from early on was not due to this baseline comparison fact (besides you compared with other state-of-the-art related methods), but due to the fact that I personally found the contribution to be (on the one hand *interesting* but on the other hand) limited from a novelty perspective.

---

> > > ### Author Response · Authors · 2018-12-10
> > > **Re: thanks for your response**
> > >
> > > Dear reviewer,
> > >
> > > Thank you for clarifying your review and reconsidering and upgrading your score!
> > >
> > > We would like to point out that Laplacian feature propagation is just that very basic PPR-based baseline you wanted to see -- it uses PPR-like feature propagation in combination with logistic regression.
> > >
> > > Since we both agree that LASAGNE falls into a different category of methods and that we use PPR in a very different way (to propagate information instead of sampling contexts for a skip-gram model), we are not quite sure what work you are referring to that reduces the novelty value of our method. Our model's simplicity might make it seem like a minor contribution but it also makes the model easy to implement, train, optimize, extend and scale. E.g. note that GNNs with many layers suffer from difficulties in gradient-based training, while our method (thanks to the decoupling of the propagation step) does not, making it more suitable to use in practice.

---

### Author Response · Authors · 2018-11-13
**Title and name change**

Dear reviewers, dear commenters,
We feel that the term "embedding" that we used in our work (and paper’s title) might be a source of confusion, which is why we have decided to replace it with “prediction” and rename the model. We want to clarify that we do NOT learn individual node embeddings as done in node embedding methods. We propagate the predictions as part of the end-to-end trained model. Please keep in mind that we did NOT change any part of the model except for the name.

---

### Public Comment · ~Benedek_Rozemberczki1 · 2019-02-02
**Code**

Is there publicly available code for the paper?

---

> ### Author Response · Authors · 2019-02-03
> **Later this month**
>
> We're planning to release source code for the model along with the camera ready version later this month.

---

> > ### Public Comment · ~Benedek_Rozemberczki1 · 2019-02-20
> > **Attempt to reproduce results.**
> >
> > I tried to reproduce the results and created an implementation.
> >
> > https://github.com/benedekrozemberczki/APPNP

---

> > > ### Author Response · Authors · 2019-02-20
> > > **Reference implementation published**
> > >
> > > Thank you for your interest and effort in reimplementing our model in PyTorch!
> > >
> > > We've just published a reference implementation at https://github.com/klicperajo/ppnp .

---

> > > > ### Public Comment · ~Benedek_Rozemberczki1 · 2019-02-20
> > > > **Thank You!**
> > > >
> > > > Referenced it on the Pytorch github repo. Loved this paper.

---

### Meta-Review · Area_Chair1 · 2018-12-14
**borderline paper**

**Confidence:** 3
**Recommendation:** Accept (Poster)

**Metareview:**

There were several ambivalent reviews for this submission and one favorable one. Although this is a difficult case, I am recommending accepting the paper.

There were two main questions in my mind.
1. Did the authors justify that the limited neighborhood problem they try to fix with their method is a real problem and that they fixed it? If so, accept.

Here I believe evidence has been presented, but the case remains undecided.

2. If they have not, is the method/experiments sufficiently useful to be interesting anyway?

This question I would lean towards answering in the affirmative.

I believe the paper as a whole is sufficiently interesting and executed sufficiently well to be accepted, although I was not convinced of the first point (1) above. One review voting to reject did not find the conceptual contribution very valuable but still thought the paper was not severely flawed. I am partly down-weighting the conceptual criticism they made. I am more concerned with experimental issues. However, I did not see sufficiently severe issues raised by the reviewers to justify rejection.

Ultimately, I could go either way on this case, but I think some members of the community will benefit from reading this work enough that it should be accepted.